# Predictors of Knowledge, Attitudes and Practice Regarding Heat Waves: An Exploratory Cross-Sectional Study in Greece

Ioannis Moisoglou [1,*], Aglaia Katsiroumpa [2], Antigoni Kolisiati [3], Evangelia Meimeti [4], Ioanna Prasini [5], Maria Tsiachri [6], Olympia Konstantakopoulou [7], Parisis Gallos [2] and Petros Galanis [2]

1. Faculty of Nursing, University of Thessaly, 41500 Larissa, Greece
2. Clinical Epidemiology Laboratory, Faculty of Nursing, National and Kapodistrian University of Athens, 11527 Athens, Greece; aglaiakat@nurs.uoa.gr (A.K.); parisgallos@nurs.uoa.gr (P.G.); pegalan@nurs.uoa.gr (P.G.)
3. Department of Endocrinology and Diabetes Center, General Hospital "G. Gennimatas", 11527 Athens, Greece; antigonikol@outlook.com
4. 3rd Regional Health Authority of Macedonia, 54630 Thessaloniki, Greece; phd_koinotiki@3ype.gr
5. Palliative Care Unit Galilee, 19004 Spata, Greece; iprasini@galilee.gr
6. Primary Education Directorate of Fthiotida, 35131 Lamia, Greece; tsiachrima@sch.gr
7. Center for Health Services Management and Evaluation, Faculty of Nursing, National and Kapodistrian University of Athens, 11527 Athens, Greece; olykonstant@nurs.uoa.gr
* Correspondence: iomoysoglou@uth.gr; Tel.: +30-2231029188

**Abstract:** Heat waves are a significant consequence of climate change, threatening public health by increasing morbidity and mortality. The aim of this study was to estimate individuals' knowledge, attitudes and practice related to heat waves. We conducted an exploratory cross-sectional study in Greece during September 2023. We employed a convenience sample of 1055 participants. We used the heat wave knowledge, awareness, practice and behavior scale (HWKAPBS) to measure our outcome. We measured several socio-demographic variables, such as gender, age and educational level, as potential determinants. Mean scores for the knowledge, awareness, practice and behavior factors were 12.5, 22.7, 22.2 and 12.1, respectively. Females had higher scores for the four factors compared with males. We found a positive relationship between self-perceived health status and awareness, practice and behavior concerning heat waves. Similarly, we identified a positive relationship between self-perceived financial status, and awareness and behavior concerning heat waves. Increased age was associated with an increased practice score, while increased educational level was associated with an increased knowledge score. Additionally, the behavior score was higher among participants in urban areas than those in rural areas. We found statistically significant positive correlations between the four factors. Levels of knowledge, awareness, practice and behavior concerning heat waves were high in our sample. Several socio-demographic variables affect participants' knowledge, awareness, practice and behavior concerning heat waves.

**Keywords:** heat waves; climate change; knowledge; attitudes; practice; behavior

## 1. Introduction

Climate change can be defined as long-term changes in temperatures and weather patterns [1]. The consequences of climate change affect the natural world, and climate change poses risks to social threats and business activities. In particular, with regard to the natural world, some of the most significant impacts of climate change include global warming; an increase in the frequency, intensity and duration of droughts; an increase in precipitation in many areas; rising sea levels; and changes in biodiversity [2]. Regarding societal changes and risks to business, these may include deaths from very high or very low temperatures and natural disasters (e.g., floods, storms and droughts), emerging and re-emerging animal diseases that present animal and human health challenges, and significant impacts on agricultural production and tourism [2–5].



Global warming, as one of the most important consequences of climate change, is defined as "the increase in the surface average temperature of the earth" [6]. The occurrence of heat waves has a direct correlation with global warming. In particular, the duration, intensity and recurrence intervals of the heat waves are affected by even a small increase in global warming [7]. Global warming is an ongoing phenomenon that has been particularly intense in recent years, as the year 2022 was the sixth warmest year in the history of global temperature records, which began in 1880 [8]. If global warming continues at the current rate, we should expect more and more heat waves in the coming years, which both governments through health systems and citizens themselves will have to prepare to respond to. Among the regions with a high incidence of heat wave phenomena is Southern Europe and, in particular, the Mediterranean, where increased heat wave days, maximum heat wave duration, average heat wave intensity and cumulative heat have been recorded over time [9].

The impacts of heat waves are multifaceted, affecting the natural environment, the animal kingdom and people. Heat waves have resulted in a significant loss of cropland, which in Europe, has tripled in recent decades [10]; an increase in electricity demand and damage to urban infrastructure [11]; devastating impacts on marine ecosystems with a reduction in the abundance of habitat-forming seaweeds; and the increased mortality of several marine species [12]. In humans, as high temperatures affect almost all organs and bodily systems, the risk of morbidity and mortality is increased, especially for patients with chronic diseases [13]. A study in the UK showed that the incidence of respiratory deaths was more pronounced during hotter weather, and the existence of a correlation (less consistent) was discovered between heat and other health outcomes such as hospital admissions, myocardial infarctions and birth outcomes [14]. Other population groups that are vulnerable in the event of a heat wave are the elderly, females, persons with lower educational attainment and individuals of a low socioeconomic status [15,16]. Infants, particularly neonates, as well as outdoor workers and those who use alcohol, medications and illegal narcotics are at a high risk of heat wave mortality [13]. Heat waves also affect people's mental health. During these periods of heat waves, there is an increase in suicides, as well as increased risks of mental health-related admissions and emergency department visits [17]. Finally, heat waves affect health services both in terms of equipment and in terms of increased demand for health services (ambulances, general practitioners, emergency services) [18,19].

Although the role of public awareness in health-protective behaviors is fundamental to reducing the risks of heat waves [20], the existence of studies using a validated tool to assess citizens' knowledge and attitudes towards heat waves is very limited. Therefore, it is necessary to recognize the factors that influence the general public's knowledge, attitudes and practice concerning heat waves. Recently, two valid instruments have been developed to measure knowledge, attitudes and practice concerning heat waves in the general population [21,22]. However, according to our knowledge, there are no studies that investigate the predictors of knowledge, attitudes and practice concerning heat waves by using valid instruments.

Thus, the objectives of our study were to (a) assess participants' knowledge, attitudes and practice related to heat waves, and (b) identify predictors of knowledge, attitudes and practice related to heat waves by using a valid tool.

## 2. Materials and Methods

### 2.1. Study Design

Since our knowledge regarding the determinants of individuals' knowledge, attitudes and practice related to heat waves is limited, we conducted an exploratory cross-sectional study in Greece. We collected our data during September 2023, after one of the hottest summers in human history, following an intense and prolonged series of reported heat waves in Europe [23]. According to a recent estimate, Greece was amongst the European countries with the highest summer heat-related mortality rate between June and August

2022 [24]. In general, the highest heat-related mortality rates were found in Mediterranean countries, such as, Greece, Italy, Spain and Portugal. Moreover, mean temperature in Greece during 2021 was amongst the fourth highest values in European countries [25].

We used several sources to collect our data. In particular, we created an online form of a study questionnaire using Google Forms and posted it on social media (Facebook, Instagram, LinkedIn, Viber and WhatsApp). Moreover, we sent the questionnaire to our e-mail contacts asking recipients to forward it to their contacts. Snowball sampling was therefore applied. By applying these online methods of data collection, we obtained data from residents from several regions throughout the country. Additionally, we approached individuals in person to request them to fill out our questionnaire. Specifically, we approached individuals in public areas of Athens, such as squares and churches. Overall, a convenience sample was obtained.

Our eligibility criteria were as follows: age $\geq$ 18 years, an understanding of the Greek language since the study questionnaire was in Greek, and a written consent to participate in our study.

Considering the low effect size ($f^2 = 0.02$) of each socio-demographic variable on each construct score, the number of predictors (n = 10), level of alpha error of 5%, level of power of 95% and a two-tailed test, we needed 652 participants to perform multivariable linear regression analysis. We therefore decided to further increase our sample to reduce random error.

*2.2. Measures*

The heat wave knowledge, awareness, practice and behavior scale (HWKAPBS) created from Sayili et al. in Turkey was used [21]. Another, similar tool has been created by scholars in Malaysia [22]. We decided to use the HWKAPBS for two reasons: (a) Turkey is a Mediterranean country near Greece and so climate conditions are similar, (b) the tool has already been validated in Greek [26] and is proven to be valid and reliable. The HWKAPBS measured knowledge about heat waves (15 items), awareness (five items), practice (five items) and behavior (three items). Correct answers take a value of 1, while wrong answers take a value of 0. Thus, an overall knowledge score is estimated with values from 0 to 15. Answers on awareness, practice and behavior factors are on a five-point Likert scale: strongly disagree (1), disagree (2), not sure (3), agree (4), strongly agree (5). We calculated an overall score for each factor by adding answers. Thus, awareness and practice scores range from 5 to 25, while behavior score ranges from 3 to 15. Higher values in the four factors of the HWKAPBS indicate higher levels of knowledge, awareness, practice and behavior regarding heat waves. In our study, Cronbach's alpha for the knowledge, awareness, practice and behavior factors were 0.72, 0.95, 0.90 and 0.80, respectively.

We collected data regarding the following socio-demographic variables: gender (females or males), age (continuous variable), educational level (elementary school, high school, university degree, MSc/PhD diploma), living conditions (alone or with others), residence area (urban or rural), workers (no or yes), air-conditioner ownership (no or yes), voluntary activities (no or yes), self-perceived health status and self-perceived financial status. We used a 5-point Likert scale to measure self-perceived health status and financial status: very poor (1), poor (2), moderate (3), good (4), very good (5). We considered these socio-demographic variables as potential predictors of knowledge, awareness, practice and behavior regarding heat waves.

*2.3. Ethical Issues*

Approval from the Ethics Committee of the Faculty of Nursing, National and Kapodistrian University of Athens was obtained to perform our study (reference number, 459; September 2023). Additionally, we applied the guidelines of the Declaration of Helsinki in our study [27]. Furthermore, participants gave their written consent to participate in our study, while personal data were not collected.

### 2.4. Statistical Analysis

Categorical variables are presented with numbers and percentages. Moreover, continuous variables are presented using mean, standard deviation (SD), median, minimum value and maximum value. Kolmogorov–Smirnov test and Q-Q plots were used to identify the distribution of continuous variables. We found that continuous variables followed normal distribution. Since knowledge, awareness, practice and behavior scores followed normal distribution, linear regression analysis was used to estimate the independent effect of socio-demographic characteristics. First, univariate linear regression analysis was performed and then multivariable linear regression models were constructed to eliminate confounding. In regression analyses, socio-demographic characteristics were the independent variables, while knowledge, awareness, practice and behavior scores were the dependent variables. Specifically, we conducted univariate linear regression analysis between each independent variable and each dependent variable to assess the relationship between these variables. Then, all independent variables were included in a final multivariable model to assess the net effect of each independent variable on the dependent variables by eliminating confounding caused by the other independent variables. Since there were four dependent variables, (i.e., knowledge, awareness, practice and behavior scores), four multivariable linear regression models were constructed. We present unadjusted and adjusted coefficient betas, 95% confidence intervals (CIs), *p*-values and coefficients of determination ($R^2$). *p*-values less than 0.05 were considered statistically significant. IBM SPSS 21.0 (IBM Corp., IBM SPSS Statistics for Windows, Version 21.0. Armonk, NY, USA, released 2012) was utilized to perform our statistical analysis.

### 3. Results

#### 3.1. Socio-Demographic Characteristics

The study population included 1055 participants. The mean age was 35.9 years (SD: 12.2). The majority of participants were females (78.6%), had a university degree (86.1%), were working (83.1%) and were living with others (82.8%) in urban areas (87.6%). Nine out of ten participants owned an air-conditioner (89.7%). Among our participants, 14.1% have participated in voluntary activities. Only 3.5% considered their health status as poor/very poor, while 84.0% considered it as good/very good. Regarding financial status, 50.9% reported a good/very good status, 43.4% reported a moderate level and 5.7% reported a poor/very poor status. Detailed socio-demographic characteristics of our sample are shown in Table 1.

**Table 1.** Socio-demographic characteristics of our sample (N = 1055).

| Characteristics | N | % |
|---|---|---|
| Gender | | |
| Females | 829 | 78.6 |
| Males | 226 | 21.4 |
| Age [a] | 35.9 | 12.2 |
| Educational level | | |
| High school | 147 | 13.9 |
| University degree | 406 | 38.5 |
| MSc diploma | 405 | 38.4 |
| PhD diploma | 97 | 9.2 |
| Living | | |
| Alone | 181 | 17.2 |
| With others | 874 | 82.8 |
| Residence area | | |
| Rural | 131 | 12.4 |
| Urban | 924 | 87.6 |

**Table 1.** *Cont.*

| Characteristics | N | % |
|---|---|---|
| Workers | | |
| No | 178 | 16.9 |
| Yes | 877 | 83.1 |
| Air-conditioner ownership | | |
| No | 109 | 10.3 |
| Yes | 946 | 89.7 |
| Voluntary activities | | |
| No | 906 | 85.9 |
| Yes | 149 | 14.1 |
| Self-perceived health status | | |
| Very poor | 25 | 2.4 |
| Poor | 12 | 1.1 |
| Moderate | 132 | 12.5 |
| Good | 407 | 38.6 |
| Very good | 479 | 45.4 |
| Self-perceived financial status | | |
| Very poor | 8 | 0.8 |
| Poor | 52 | 4.9 |
| Moderate | 458 | 43.4 |
| Good | 472 | 44.7 |
| Very good | 65 | 6.2 |

[a] mean, standard deviation.

### 3.2. Study Scales

Descriptive statistics regarding knowledge, awareness, practice and behavior factors are shown in Table 2. The mean knowledge score was 12.5 (SD: 1.9), indicating a high knowledge level in our sample. Moreover, the level of awareness of heat waves was high since the mean score was 22.7 (SD: 4.2). Participants' practices in dealing with heat waves were very good with a mean practice score of 22.2 (SD: 4.3). The high mean behavior score (12.1, SD: 2.9) indicated that participants are alert towards heat waves. We found statistically significant positive correlations between the four factors (Table 3). The correlation between knowledge and other factors was weak since correlation coefficients ranged from 0.082 to 0.144. Moreover, correlations between awareness, practice and behaviors were strong, ranging from 0.647 to 0.738.

**Table 2.** Descriptive statistics for the knowledge, awareness, practice and behavior factors (N = 1055).

| Factor | Mean | Standard Deviation | Median | Minimum Value | Maximum Value |
|---|---|---|---|---|---|
| Knowledge | 12.5 | 1.9 | 13 | 3 | 15 |
| Awareness | 22.7 | 4.2 | 24 | 5 | 25 |
| Practice | 22.2 | 4.3 | 24 | 5 | 25 |
| Behavior | 12.1 | 2.9 | 13 | 3 | 15 |

**Table 3.** Pearsons' correlation coefficients between the knowledge, awareness, practice and behavior factors (N = 1055).

| Factor | 2 | 3 | 4 |
|---|---|---|---|
| 1. Knowledge | 0.144 * | 0.082 ** | 0.086 ** |
| 2. Awareness | | 0.738 * | 0.647 * |
| 3. Practice | | | 0.725 * |
| 4. Behavior | | | |

* $p$-value < 0.001, ** $p$-value < 0.01.

*3.3. Regression Analysis*

Females (adjusted beta: 0.43, 95% CI: 0.14 to 0.71) and owners of an air-conditioner (adjusted beta: 0.71, 95% CI: 0.33 to 1.10) had more knowledge about heat waves (Table 4). Moreover, we found a positive relationship between educational level and knowledge score. Specifically, participants with a university degree (adjusted beta: 0.44, 95% CI: 0.08 to 0.80), an MSc degree (adjusted beta: 0.74, 95% CI: 0.37 to 1.12), or a PhD degree (adjusted beta: 0.88, 95% CI: 0.37 to 1.40) were more knowledgeable compared to high school graduates. Additionally, the knowledge score was higher among participants with good/very good self-perceived health status compared with those with poor/very poor health status (adjusted beta: 0.69, 95% CI: 0.04 to 1.34).

**Table 4.** Linear regression analysis with knowledge score as the dependent variable (N = 1055).

| Independent Variables | Univariate Model | | Multivariable Model | |
|---|---|---|---|---|
| | Unadjusted Coefficient Beta (95% CI) | *p*-Value | Adjusted Coefficient Beta (95% CI) | *p*-Value |
| Females vs. males | 0.36 (0.08 to 0.64) | 0.013 | 0.43 (0.14 to 0.71) | 0.003 |
| Age | −0.0001 (−0.01 to 0.01) | 0.976 | −0.006 (−0.02 to 0.004) | 0.250 |
| Educational level | | | | |
| University degree vs. high school | −0.13 (−0.37 to 0.11) | 0.277 | 0.44 (0.08 to 0.80) | 0.018 |
| MSc degree vs. high school | 0.37 (0.13 to 0.61) | 0.002 | 0.74 (0.37 to 1.12) | <0.001 |
| PhD degree vs. high school | 0.30 (−0.10 to 0.70) | 0.144 | 0.88 (0.37 to 1.40) | 0.001 |
| Living with other | −0.04 (−0.35 to 0.27) | 0.799 | 0.06 (−0.25 to 0.36) | 0.728 |
| Urban vs. rural residence area | 0.21 (−0.14 to 0.56) | 0.246 | 0.05 (−0.31 to 0.41) | 0.792 |
| Workers | 0.25 (−0.06 to 0.56) | 0.119 | 0.07 (−0.27 to 0.41) | 0.671 |
| Air-conditioner ownership | 0.73 (0.35 to 1.11) | <0.001 | 0.71 (0.33 to 1.10) | <0.001 |
| Voluntary activities | 0.17 (−0.17 to 0.50) | 0.326 | 0.10 (−0.23 to 0.44) | 0.544 |
| Self-perceived health status | | | | |
| Moderate to poor/very poor | −0.15 (−0.50 to 0.20) | 0.400 | 0.50 (−0.20 to 1.20) | 0.161 |
| Good/very good to poor/very poor | 0.29 (−0.03 to 0.61) | 0.072 | 0.69 (0.04 to 1.34) | 0.037 |
| Self-perceived financial status | | | | |
| Moderate to poor/very poor | 0.10 (−0.13 to 0.34) | 0.386 | 0.25 (−0.28 to 0.78) | 0.348 |
| Good/very good to poor/very poor | −0.01 (−0.24 to 0.23) | 0.947 | 0.11 (−0.43 to 0.65) | 0.689 |

CI: confidence interval, adjusted $R^2$ for the model = 3.4%; *p*-value for ANOVA < 0.001.

The linear regression analysis results with awareness score as the dependent variable are shown in Table 5. Mean awareness score was higher among females compared with males (adjusted beta: 2.07, 95% CI: 1.47 to 2.66). Moreover, better health status and financial status were associated with an increased awareness score. In particular, participants with moderate (adjusted beta: 4.56, 95% CI: 3.08 to 6.03) or good/very good health status (adjusted beta: 4.68, 95% CI: 3.31 to 6.05) had higher awareness scores compared to those with poor/very poor health status. Additionally, levels of awareness were higher among participants with moderate (adjusted beta: 1.12, 95% CI: 0.003 to 2.23) or good/very good financial status (adjusted beta: 2.03, 95% CI: 0.89 to 3.16).

Table 6 shows the results of the linear regression analysis with practice score as the dependent variable. Practice score was higher among females (adjusted beta: 2.03, 95% CI: 1.42 to 2.65) compared with males. Additionally, participants with moderate (adjusted beta: 5.08, 95% CI: 3.57 to 6.58) or good/very good health status (adjusted beta: 5.22, 95% CI: 3.82 to 6.61) had higher practice scores compared to those with poor/very poor health status. Moreover, we found a positive relationship between age and practice score (adjusted beta: 0.04, 95% CI: 0.02 to 0.06).

**Table 5.** Linear regression analysis with awareness score as the dependent variable (N = 1055).

| Independent Variables | Univariate Model | | Multivariable Model | |
|---|---|---|---|---|
| | Unadjusted Coefficient Beta (95% CI) | *p*-Value | Adjusted Coefficient Beta (95% CI) | *p*-Value |
| Females vs. males | 1.80 (1.19 to 2.41) | <0.001 | 2.07 (1.47 to 2.66) | <0.001 |
| Age | −0.02 (−0.04 to 0.0005) | 0.055 | −0.02 (−0.04 to 0.0001) | 0.051 |
| Educational level | | | | |
| University degree vs. high school | −0.51 (−1.03 to 0.01) | 0.055 | −0.26 (−1.02 to 0.50) | 0.506 |
| MSc degree vs. high school | 0.49 (−0.03 to 1.02) | 0.065 | 0.56 (−0.23 to 1.35) | 0.165 |
| PhD degree vs. high school | 0.49 (−0.39 to 1.38) | 0.271 | 0.99 (−0.09 to 2.07) | 0.072 |
| Living with other | −0.11 (−0.78 to 0.57) | 0.760 | −0.002 (−0.65 to 0.65) | 0.996 |
| Urban vs. rural residence area | 0.33 (−0.44 to 1.10) | 0.399 | 0.43 (−0.32 to 1.19) | 0.260 |
| Workers | −0.38 (−1.06 to 0.30) | 0.277 | −0.60 (−1.31 to 0.12) | 0.102 |
| Air-conditioner ownership | 0.70 (−0.14 to 1.54) | 0.100 | 0.78 (−0.03 to 1.60) | 0.059 |
| Voluntary activities | −0.16 (−0.89 to 0.57) | 0.659 | −0.28 (−0.99 to 0.42) | 0.431 |
| Self-perceived health status | | | | |
| Moderate to poor/very poor | −0.30 (−1.07 to 0.47) | 0.443 | 4.56 (3.08 to 6.03) | <0.001 |
| Good/very good to poor/very poor | 1.51 (0.82 to 2.20) | <0.001 | 4.68 (3.31 to 6.05) | <0.001 |
| Self-perceived financial status | | | | |
| Moderate to poor/very poor | −0.69 (−1.20 to −0.18) | 0.008 | 1.12 (0.003 to 2.23) | 0.049 |
| Good/very good to poor/very poor | 1.19 (0.69 to 1.70) | <0.001 | 2.03 (0.89 to 3.16) | <0.001 |

CI: confidence interval, adjusted $R^2$ for the model = 10.9%; *p*-value for ANOVA < 0.001.

**Table 6.** Linear regression analysis with practice score as the dependent variable (N = 1055).

| Independent Variables | Univariate Model | | Multivariable Model | |
|---|---|---|---|---|
| | Unadjusted Coefficient Beta (95% CI) | *p*-Value | Adjusted Coefficient Beta (95% CI) | *p*-Value |
| Females vs. males | 1.75 (1.12 to 2.37) | <0.001 | 2.03 (1.42 to 2.65) | <0.001 |
| Age | 0.03 (0.01 to 0.06) | 0.001 | 0.04 (0.02 to 0.06) | 0.001 |
| Educational level | | | | |
| University degree vs. high school | −0.88 (−1.41 to −0.35) | 0.001 | −0.26 (−1.03 to 0.52) | 0.518 |
| MSc diploma vs. high school | 1.04 (0.51 to 1.57) | <0.001 | 0.82 (−0.01 to 1.63) | 0.052 |
| PhD diploma vs. high school | 0.23 (−0.67 to 1.13) | 0.616 | 0.46 (−0.64 to 1.57) | 0.410 |
| Living with other | −0.22 (−0.90 to 0.47) | 0.537 | −0.22 (−0.88 to 0.45) | 0.524 |
| Urban vs. rural residence area | 0.16 (−0.62 to 0.95) | 0.683 | 0.23 (−0.54 to 1.003) | 0.552 |
| Workers | 0.15 (−0.54 to 0.85) | 0.662 | −0.68 (−1.41 to 0.05) | 0.066 |
| Air-conditioner ownership | 0.79 (−0.06 to 1.64) | 0.067 | 0.83 (−0.0005 to 1.66) | 0.050 |
| Voluntary activities | −0.34 (−1.08 to 0.41) | 0.375 | −0.49 (−1.21 to 0.23) | 0.183 |
| Self-perceived health status | | | | |
| Moderate to poor/very poor | −0.04 (−0.82 to 0.75) | 0.930 | 5.08 (3.57 to 6.58) | <0.001 |
| Good/very good to poor/very poor | 1.28 (0.58 to 1.98) | <0.001 | 5.22 (3.82 to 6.61) | <0.001 |
| Self-perceived financial status | | | | |
| Moderate to poor/very poor | −0.35 (−0.87 to 0.17) | 0.189 | 0.65 (−0.49 to 1.79) | 0.261 |
| Good/very good to poor/very poor | 0.76 (0.24 to 1.27) | 0.004 | 1.04 (−0.12 to 2.20) | 0.080 |

CI: confidence interval, adjusted $R^2$ for the model = 10.5%; *p*-value for ANOVA < 0.001.

We present the results of the linear regression analysis with behavior score as the dependent variable in Table 7. We found that females had better behavior towards heat waves than males (adjusted beta: 1.17, 95% CI: 0.74 to 1.60). Additionally, behavior score was higher among participants in urban areas than those in rural areas (adjusted beta: 0.62, 95% CI: 0.08 to 1.16). Moreover, increased health status and financial status were associated with an increased behavior score. In particular, participants with moderate (adjusted beta: 2.38, 95% CI: 1.33 to 3.44) or good/very good health status (adjusted beta: 1.78, 95% CI: 0.80 to 2.76) had higher behavior scores compared to those with poor/very poor health status. Also, participants with moderate (adjusted beta: 0.90, 95% CI: 0.10 to 1.70) or good/very

good financial status (adjusted beta: 1.70, 95% CI: 0.89 to 2.51) had higher behavior scores compared to those with poor/very poor health status.

**Table 7.** Linear regression analysis with behavior score as the dependent variable (N = 1055).

| Independent Variables | Univariate Model | | Multivariable Model | |
|---|---|---|---|---|
| | Unadjusted Coefficient Beta (95% CI) | *p*-Value | Adjusted Coefficient Beta (95% CI) | *p*-Value |
| Females vs. males | 1.02 (0.59 to 1.44) | <0.001 | 1.17 (0.74 to 1.60) | <0.001 |
| Age | 0.001 (−0.01 to 0.02) | 0.934 | 0.001 (−0.02 to 0.02) | 0.923 |
| Educational level | | | | |
| University degree vs. high school | −0.11 (−0.48 to 0.25) | 0.544 | −0.17 (−0.71 to 0.37) | 0.543 |
| MSc degree vs. high school | 0.08 (−0.29 to 0.44) | 0.683 | −0.05 (−0.61 to 0.52) | 0.872 |
| PhD degree vs. high school | 0.20 (−0.41 to 0.82) | 0.519 | 0.10 (−0.68 to 0.87) | 0.807 |
| Living with other | −0.04 (−0.51 to 0.43) | 0.864 | −0.02 (−0.48 to 0.45) | 0.944 |
| Urban vs. rural residence area | 0.50 (−0.04 to 1.04) | 0.067 | 0.62 (0.08 to 1.16) | 0.024 |
| Workers | −0.13 (−0.61 to 0.34) | 0.584 | −0.18 (−0.70 to 0.33) | 0.479 |
| Air-conditioner ownership | 0.11 (−0.47 to 0.69) | 0.711 | 0.04 (−0.54 to 0.63) | 0.882 |
| Voluntary activities | −0.25 (−0.76 to 0.26) | 0.339 | −0.29 (−0.80 to 0.21) | 0.253 |
| Self-perceived health status | | | | |
| Moderate to poor/very poor | 0.38 (−0.15 to 0.92) | 0.160 | 2.38 (1.33 to 3.44) | <0.001 |
| Good/very good to poor/very poor | 0.24 (−0.25 to 0.72) | 0.341 | 1.78 (0.80 to 2.76) | <0.001 |
| Self-perceived financial status | | | | |
| Moderate to poor/very poor | −0.52 (−0.88 to −0.16) | 0.004 | 0.90 (0.10 to 1.70) | 0.027 |
| Good/very good to poor/very poor | 0.82 (0.47 to 1.17) | <0.001 | 1.70 (0.89 to 2.51) | <0.001 |

CI: confidence interval, adjusted $R^2$ for the model = 6.2%; *p*-value for ANOVA < 0.001.

## 4. Discussion

The present study assessed participants' knowledge, attitudes, practice and behavior related to heat waves and investigated predictors of these variables. According to the study results, participants were found to have a high level of knowledge, attitudes, behavior and practice regarding heat waves, as they scored high in all subscales. Greece is located in the Mediterranean region, where heat waves of particularly high intensity, frequency and duration are common [9]. The experiences of residents, combined with the information provided by the state, may have contributed in improving the level of knowledge and public awareness in health-protective behaviors in the case of heat waves. A study in the United States showed that residents living in the hottest states had high heat-risk perceptions [28].

We found that females had a better level of knowledge and attitudes compared to males. This finding is consistent with study findings where females were found to be more engaged in health-protective behavior regarding heat waves compared to males [20]. Females seem to exhibit more protective behavior than males when it comes to life-threatening situations. The same was shown in the case of the COVID-19 pandemic, where females were more likely to undergo rapid tests and comply with government guidelines than males [29,30]. Participants with a higher educational level were found to have a better level of knowledge. This finding is consistent with similar studies in the USA and Europe, where a lower educational level has been shown to be a predictor for lower knowledge levels on heat wave protective measures [31,32]. High educational attainment also plays a protective role by reducing the risk of heat wave mortality [33,34]. This finding can be explained by the fact that people with higher educational attainment may have better access to knowledge or information on heat wave health risks, vulnerable groups and protective measures. It is also possible that educational level is related to the type of jobs of individuals, as those with lower educational levels may be working outdoors and may therefore be at higher risk in the event of a heat wave.

The health status of the participants in this study was found to be a predictor of their level of knowledge, awareness, practice and behavior. Specifically, better health status was

associated with increased score in all subscales. Similar findings were made in another study, where the self-rated health was significantly associated with awareness of heat wave alerts and engagement in health-protective behaviors [20]. The advanced level of knowledge of healthier participants, regarding heat waves, may be due to their general health attitudes towards issues that can negatively affect their health. They probably choose to be informed about health and public health issues and choose healthier behaviors and lifestyles. Although citizens with a poor health status are vulnerable to heat waves [35], they are often unaware of this vulnerability and are also unaware of the effectiveness of important protective behaviors [36]. Therefore, the need for the state to intervene with targeted information programs on the risks of heat waves to vulnerable groups, such as citizens with chronic diseases and generally poor health status, is considered extremely important. The intervention programs contribute to education, awareness raising and the development of adaptive behaviors to deal with the risk of heat waves [37]. It is important to educate citizens, as their perceptions of heat waves and the risks arising from them can mediate individual behavioral intentions upon exposure to high ambient temperatures [38].

Participants' economic status was found to influence awareness, practice and behaviors regarding heat waves, as it was shown that the higher the economic status, the higher the score in the relevant subscales. Our findings are confirmed by several studies; specifically, a study in New York found that low-income participants were less likely to be aware of heat warnings [39]; another study in China showed that high-income participants had higher scores in the practice section [40]; and finally, a study in Australia found that high-income participants were more likely to have good adaptive behaviors during a heat wave [41]. It is likely that high-income citizens have better access to information sources, more free time to participate in information programs or that the communities they live in are more aware of public health issues such as heat wave protection.

Our study had several limitations. Since we obtained a convenience sample, results cannot be generalized. The sample was not representative of the source population. For example, most of our participants were highly educated females. Moreover, the majority of our participants were young and middle-aged individuals living in air-conditioned apartments. Further studies with random and stratified samples can reduce our selection bias. For instance, since heat waves are a threat mainly for children and the elderly, with low income and living in apartments without air conditioning, future studies should focus on these high-risk groups. Additionally, we conducted our study in a sample of the Greek population. Therefore, we should conduct more studies in different countries, cultures and settings to further expand our conclusions. Although we investigated several socio-demographic predictors of individuals' knowledge and attitudes about heat waves, we cannot measure all potential predictors in our study. Future studies should expand our knowledge by investigating more predictors, e.g., job, psychosocial factors and personality traits. Moreover, the cross-sectional design of our study did not allow us to estimate causal relationships between socio-demographic variables and participants' knowledge and attitudes about heat waves. Longitudinal studies can reduce this bias by also assessing changes in people's behaviors through time. Additionally, we should acknowledge some pitfalls in our statistical analysis. Since our sample included highly educated individuals (i.e., all have attended at least high school), we cannot interpret our results for all education levels. Moreover, the low number of participants in some categories (e.g., individuals living in apartments without air-conditioner, very poor/poor self-perceived health and financial status) decreased the reliability of our results. Further studies with samples of greater variability can add significant knowledge on this issue. Finally, we used a self-reported scale to assess individuals' knowledge and attitudes about heat waves. Although our scale has proven to be valid and reliable in the Greek language, an information bias is probable in our study.

## 5. Conclusions

Our findings indicated a high level of knowledge and awareness about heat waves in our sample. Similarly, our participants' practices in responding to heat waves were well established. Moreover, we found strong correlations between awareness, practice and behaviors towards heat waves. Additionally, our results suggested that several socio-demographic variables affect participants' knowledge, awareness, practice and behavior concerning heat waves. Thus, our findings can be used in terms of public health protection by creating targeted awareness campaigns using the predictors that influence citizens' knowledge and behavior that the present study highlighted. Since heat waves are now increasingly common around the world, citizens' knowledge, attitudes and behaviors towards heat waves should be improved to reduce the consequences of climate change. The state should protect citizens, regardless of whether they belong to vulnerable groups, by using informative programs about potential risks and relevant precautionary measures.

**Author Contributions:** Conceptualization, P.G. (Petros Galanis) and I.M.; methodology, P.G. (Petros Galanis), I.M. and A.K. (Antigoni Kolisiati); software, P.G. (Petros Galanis) and O.K.; validation, A.K. (Aglaia Katsiroumpa), I.P., M.T., A.K. (Antigoni Kolisiati), P.G. (Parisis Gallos) and O.K.; formal analysis, P.G. (Petros Galanis) and A.K. (Aglaia Katsiroumpa); investigation, P.G. (Petros Galanis), A.K. (Aglaia Katsiroumpa), E.M., I.P., M.T., A.K. (Antigoni Kolisiati), P.G. (Parisis Gallos) and O.K.; resources, P.G. (Petros Galanis), A.K. (Aglaia Katsiroumpa), A.K. (Antigoni Kolisiati), E.M., I.P. and O.K.; data curation, I.M., A.K. (Aglaia Katsiroumpa), M.T., A.K. (Antigoni Kolisiati), P.G. (Parisis Gallos) and O.K.; writing—original draft preparation, P.G. (Petros Galanis), I.M., A.K. (Aglaia Katsiroumpa), E.M. and A.K. (Antigoni Kolisiati); writing—review and editing, P.G. (Petros Galanis), I.M., A.K. (Aglaia Katsiroumpa), E.M., I.P., M.T., A.K. (Antigoni Kolisiati), P.G. (Parisis Gallos) and O.K.; supervision, P.G. (Petros Galanis); project administration, P.G. (Petros Galanis) and I.M. All authors have read and agreed to the published version of the manuscript.

**Funding:** This research received no external funding.

**Data Availability Statement:** The data presented in this study are available on request from the corresponding author.

**Acknowledgments:** We acknowledge all the participants who made this study possible.

**Conflicts of Interest:** The authors declare no conflicts of interest.

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
