# Peer review of "Predictors of Knowledge, Attitudes and Practice Regarding Heat Waves: An Exploratory Cross-Sectional Study in Greece"

_climate, doi:10.3390/cli12030036_

Round 1

Reviewer 1 Report

Comments and Suggestions for Authors

The biggest reservations in the presented article concern the selection of respondents. The most at risk are children and the elderly, with low income and living in apartments without air conditioning. However, the participants were mainly young and middle-aged people living in air-conditioned apartments and assessing their health and finances positively. This selection makes the obtained reports unreliable.

The second problem is created by linear regression between elements, at least one
of which has only a few possible values: female or male, or air conditioner owner
(yes or no). One age can be treated as a continuous variable. The regression
calculation method used should be described in much more detail. This version
is not robust. Additionally, by calculating the regression only for a shortened range
of the independent variable (level of education starts with high school), we cannot
interpret the results for the entire range of education. Generally better description
how the univariate and multivariate models were built should be given.
  Minor comments:

Line 53: Shouldn’t it be “floods” instead of “foods”? Comments on the Quality of English Language

Only minor editing is required

Author Response

Dear Reviewer 1,

Thank you very much for the peer review of the paper “Predictors of knowledge, attitudes and practice about heat waves: An exploratory cross-sectional study in Greece” and your comments, which have improved the quality of the manuscript.

We have addressed all the comments (highlighted in yellow) in the revised text. Please, find below an item-by-item answer to your comments. Also, we made changes in the manuscript according to the other Reviewers’ instructions.

Hoping the revised manuscript fulfils the journal’s standards, we thank you for your courtesy.

We are looking forward to your response.

Yours sincerely,

The authors

Comment

The biggest reservations in the presented article concern the selection of respondents. The most at risk are children and the elderly, with low income and living in apartments without air conditioning. However, the participants were mainly young and middle-aged people living in air-conditioned apartments and assessing their health and finances positively. This selection makes the obtained reports unreliable.

Answer: Done

We recognize this limitation of our study. We further emphasize this pitfall in the Limitations section by adding the following text.

Moreover, the majority of our participants were young and middle-aged individuals living in air-conditioned apartments. Further studies with random and stratified samples can reduce our selection bias. For instance, since heat waves are a threat mainly for children and elderly, with low income and living in apartments without air conditioning future studies should focus on these high risk groups.

Comment

The second problem is created by linear regression between elements, at least one
of which has only a few possible values: female or male, or air conditioner owner
(yes or no). One age can be treated as a continuous variable. The regression
calculation method used should be described in much more detail. This version
is not robust. Additionally, by calculating the regression only for a shortened range
of the independent variable (level of education starts with high school), we cannot
interpret the results for the entire range of education. Generally better description
how the univariate and multivariate models were built should be given.   

Answer: Done

We better describe the univariate and multivariate models by adding the following text in Statistical analysis section.

…In particular, we conducted univariate linear regression analysis between each independent variable and each dependent variable to assess relationship between these variables. Then, we included all independent variables in a final multivariable model to assess the net effect of each independent variable on dependent variable by eliminating confounding caused by the other independent variables. Since we had four dependent variables, (i.e., knowledge, awareness, practice and behavior scores) we constructed four multivariable linear regression models…

Regarding educational level, all participants have attended at least high school. We did not select in our study design to include only highly educated participants. It was just happened in our study. Moreover, we recognize that unfortunately some variables have only a few possible values in our study. After your insightful comments we add these limitations in our Limitations section, by adding the following text.

…Furthermore, we should notify some pitfalls in our statistical analysis. Since our sample included highly educated individuals (i.e., all have attended at least high school) we cannot interpret our results for the entire range of education. Moreover, low number of participants in some categories (e.g. individuals living in apartments without air-conditioner, very poor/poor self-perceived health and financial status) decreased reliability of our results. Further studies with samples of greater variability can add significant knowledge on this issue…

Minor comments:

 Line 53: Shouldn’t it be “floods” instead of “foods”?

Answer: Done

We fix this error.

Reviewer 2 Report

Comments and Suggestions for Authors

Please find my comments and suggestions in the attached file.

Author Response

Dear Reviewer 2,

Thank you very much for the peer review of the paper “Predictors of knowledge, attitudes and practice about heat waves: An exploratory cross-sectional study in Greece” and your comments, which have improved the quality of the manuscript.

We have addressed all the comments (highlighted in yellow) in the revised text. Please, find below an item-by-item answer to your comments. Also, we made changes in the manuscript according to the other Reviewers’ instructions.

Hoping the revised manuscript fulfils the journal’s standards, we thank you for your courtesy.

We are looking forward to your response.

Yours sincerely,

The authors

Comment

The authors should significantly modify the introduction. As it is now, it does not reveal the relevance of this study and does not show what has already been done on the subject of the paper and what needs further research - several paragraphs should be added to show this. Instead, there are paragraphs in the introduction that are not relevant to the topic of the paper and should be deleted. For example, in lines 37-46 - it is unclear why the authors are writing about sources of greenhouse gas emissions here. The text from lines 55-62 is also irrelevant to the topic of the study and should be deleted.

Answer: Done

Lines 37-47 removed.

The text in lines 55-62, was modified to make it more relevant. As the phenomenon of global warming is related to the occurrence of heat waves, the text was better presented in order to have a logical continuity in the text and a connection to our topic.

Comment

Line 105 the authors write: "Athens is the capital and the biggest city of Greece". This is well-known information and there is no need to write it.

Answer: Done

We removed the text "Athens is the capital and the biggest city of Greece".

Comment

The authors should rewrite the conclusions, as this is one of the most important parts of the paper. The conclusions in their current form do not correspond to the results of the paper. In lines 321-322, the authors write: "Climate change has affected the temperature of the planet, as it is increasing in every region of the world. Rising temperatures are associated with the occurrence of heat waves". But this cannot be the conclusion of this paper, because the authors did not investigate the temperature rise. In lines 323-326, the authors write: "As heat waves are becoming more frequent, intense and long-lasting, it is essential for the protection of public health that citizens are aware of the risks arising from this phenomenon, the vulnerable groups in the event of heat waves and the protective measures to be taken", but this is also not the conclusion of this study - these are common knowledge. The Conclusions section does not contain any real conclusions drawn from the results of this study.

Answer: Done

We rewrite the conclusion section in order to present the value of the findings of our study.

Round 2

Reviewer 2 Report

Comments and Suggestions for Authors

The authors should modify the conclusions. The conclusions in their current form do not correspond to the results of the paper. In lines 344–347 the authors write well-known things that are not the conclusions of this study. In lines 350–351 the authors write where the results of the study can be used. But where are the actual conclusions of the study? 

Author Response

Comment

The authors should modify the conclusions. The conclusions in their current form do not correspond to the results of the paper. In lines 344–347 the authors write well-known things that are not the conclusions of this study. In lines 350–351 the authors write where the results of the study can be used. But where are the actual conclusions of the study?

Answer: Done

Authors modified the conclusion section, as below: Our findings indicated a high level of knowledge and awareness about heat waves in our sample. Similarly, our participants’ practices to respond to with heat waves were well established. Moreover, we found strong correlations between awareness, practice, and behaviors towards heat waves. Additionally, our results suggested that several socio-demographic variables affect participants’ knowledge, awareness, practice, and behavior concerning heat waves. Thus, our findings can be used in terms of public health protection by creating targeted awareness campaigns using the predictors that influence citizens' knowledge and behavior that the present study highlighted. Since heat waves are now increasingly common around the world, citizens’ knowledge, attitudes, and behaviors towards heat waves should be improved to reduce the consequences of climate change. The State should protect citizens, whether or not they belong to vulnerable groups, through programs to inform them of the risks and the precautionary measures they should take.
